# Dermatopathological findings of *Bothrops atrox* snakebites: A case series in the Brazilian Amazon

**Fabiane Bianca Albuquerque Barbosa**[1,2☯], **Rima de Souza Raad**[1☯], **Hiochelson Najibe Santos Ibiapina**[1,2], **Monique Freire dos Reis**[1,2,3], **Juliana Costa Ferreira Neves**[1,2], **Rosilene Viana Andrade**[2], **Thaís Pinto Nascimento**[1,2], **Fabio Francesconi Valle**[2,3], **Nicholas R. Casewell**[4], **Jacqueline Sachett**[1,2], **Marco Aurélio Sartim**[1,2,5], **Wuelton Monteiro**[1,2‡]*, **Allyson Guimarães Costa**[1,2,6‡], **Luiz Carlos Lima Ferreira**[1,2,3‡]

**1** Programa de Pós-Graduação em Medicina Tropical, Universidade do Estado do Amazonas (UEA), Manaus, Brazil, **2** Diretoria de Ensino e Pesquisa, Fundação de Medicina Tropical Doutor Heitor Vieira Dourado (FMT-HVD), Manaus, Brazil, **3** Faculdade de Medicina, Universidade Federal do Amazonas (UFAM), Manaus, Brazil, **4** Centre for Snakebite Research and Interventions, Liverpool School of Tropical Medicine, Liverpool, United Kingdom, **5** Departamento de Pesquisa, Universidade Nilton Lins (UNL), Manaus, Brazil, **6** Programa de Pós-Graduação em Imunologia Básica e Aplicada, Instituto de Ciências Biológicas, Universidade Federal do Amazonas, Manaus, Brazil

☯ These authors contributed equally to this work.
‡ These authors are joint senior authors on this work.
* wueltonmm@gmail.com

## Abstract

### Background

*Bothrops* venom consists primarily of metalloproteinase and phospholipase A2 toxins, which are responsible for the acute inflammatory, coagulant and hemorrhagic action following snakebite. The local effects of snakebite envenomation by *Bothrops* species are particularly prevalent yet poorly studied, but include pain, edema, erythema, blistering, bleeding, and ecchymosis.

### Methods and findings

In this study, we describe the dermatopathological findings observed in a series of 22 patients diagnosed with *Bothrops* envenomation treated in a tertiary hospital of Manaus, in the Brazilian Amazon. Clinically, pain and edema were observed in all patients, followed by fang marks (63.6%), secondary infection (36.3%), ecchymosis (31.8%), erythema (22.7%), blister (13.6%), and necrosis (4.5%). Regarding histopathological findings, epidermal alterations such as spongiosis, acanthosis and hyperkeratosis were the most observed characteristics in our cases series, with isolated cases of hyperplasia, hemorrhagic intraepidermal blister and severe necrosis. Changes in dermis and hypodermis consisted mainly of hemorrhage, inflammatory infiltrate, edema, congestion, and vascular damage, whereas cases of collagen damage, necrosis, abscess, and signs of tissue repair, indicated by the presence of granulation tissue, were also observed, with a persistence of inflammatory and hemostatic alterations even days after antivenom administration. Therefore, the tissue damage

available without restriction. All relevant data are within the paper.

**Funding:** Financial support was provided in the form of grants from Fundação de Amparo à Pesquisa do Estado do Amazonas - FAPEAM (Pró-Estado Program #002/2008, #007/2018 and #005/2019, to W.M.; POSGRAD Program #002/2024 and AMAZÔNIDAS Program #002/2021, to J.S.; PhD fellowships - 038/2022 - PDPG, to F.B.A.B, J.C.F.N, and T.P.N.), Conselho Nacional de Desenvolvimento Científico e Tecnológico - CNPq (Research fellowships 305762/2022-2 to A.G.C., 311434/2021-5 to J.S., and 303207/2020-7 to W.M.) and Coordenação de Aperfeiçoamento de Pessoal de Nível Superior - CAPES (PhD scholarship - #001 to H.N.S.I). The funders had no role in study design, the decision to publish, or preparation of the manuscript.

**Competing interests:** The authors have declared that no competing interests exist.

resulting from *Bothrops* envenomation could be related to both direct venom activity as well as inflammatory response or presence of infectious process. The histopathological analysis of human skin injury can enlighten the pathological and endogenous effects of local envenomation and could underpin new strategies, including novel treatments, adjuvants or changes in clinical management, that lead to better outcomes in snakebite patients.

## Author summary

After inoculation of the venom by a snake of the genus *Bothrops*, a series of changes occur around the site of the bite, as a direct effect of toxins producing tissue damage and changes in blood flow to the site associated to coagulation disorders. Current knowledge about histopathological changes resulting from snakebites is almost exclusively derived from experimental animal models. In this study, we describe the dermatopathological findings observed in a series of 22 patients diagnosed with *Bothrops atrox* envenomation treated in a tertiary hospital of Manaus, in the Brazilian Amazon. We found relevant changes in all strata of the skin, with emphasis on hemorrhage, inflammatory infiltrate, edema, congestion, and vascular damage. Interestingly, we observed a persistence of inflammatory and vascular alterations days after antivenom administration, suggesting that antivenom may have limited efficacy in reversing or preventing local injury mechanisms, and that inflammation following those pathological processes plays a major role in the progression of tissue damage.

## Introduction

Snakebite envenomation results from the inoculation of toxins in the venom of snakes into humans or animals and is considered a major public health issue, mainly in countries of Asia, Africa, Oceania and Latin America [1–3]. In the Brazilian Amazon, *Bothrops* snakebites stand out due to the large number of cases reported annually, and predominately affects rural workers involved in farming, forestry and hunting [4,5].

*Bothrops* envenomation is characterized by local and systemic effects. Venom-induced consumption coagulopathy is a hallmark of systemic envenomation, resulting in ischemic and hemorrhagic manifestations [6]. Local tissue damage can occur through direct action of venom toxins on the tissue, as well as through the acute inflammatory activity promoted by such components, leading to pain and edema [7–10]. Local complications such as blisters and necrosis are relatively common and may cause permanent damage to the patient [11–13]. Furthermore, secondary bacterial infections affect 40% of *Bothrops* envenomations in the Brazilian Amazon [12].

*Bothrops* venom consists of a complex mix of toxins composed primarily of snake venom metalloproteinases (SVMPs), serine proteases (SPs), phospholipases A2 (PLA$_2$) that lead to the pathophysiological effects of envenomation [14]. MPs are the major components of *Bothrops atrox* venom and act mainly by degrading extracellular matrix components, causing damage to the basement membrane of capillaries and the dermal-epidermal junction, while others activate components of the coagulation system and trigger the release of inflammatory mediators [9,15–18].

The local tissue damage caused by *Bothrops* venom leads to important histological changes. Studies carried out in animal models showed the action of the venom in destroying muscle

tissue and peripheral nerves [19–21]. In addition, the inoculation of the venom into the skin of mice demonstrated the rapid action of the venom in the destruction of components of the dermis and dermal-epidermal layer [22]. However, studies reporting pathological findings from patients are scarce and focused on systemic effects. For example, in a case of a patient bitten by *Bothrops jararacussu*, alveolar hemorrhage and edema with multiple fibrin thrombi present in microcirculation of lungs were reported [23]. Additionally, a case series of 29 *B. jararacussu* snakebites presented, among microscopical findings, acute tubular and bilateral renal cortical necrosis, diffuse glomerulonephritis, thrombi in small arteries, arterioles and glomerular capillaries of the kidney, as well as infiltration of polymorphonuclear cells in hepatic sinusoids and portal tracts [24]. However, few studies have reported on the local effects of envenomation in snakebite patients in the region. Consequently, in this study, we describe the dermatopathological findings observed in a series of 22 patients diagnosed with *Bothrops atrox* envenomation treated in a tertiary hospital of Manaus, in the Brazilian Amazon.

## Material and methods

### Ethics statement

This study was approved by the ethics committee of the Fundação de Medicina Tropical Dr. Heitor Vieira Dourado (FMT-HVD), the reference hospital for the treatment of animal envenomation in Manaus, Western Brazilian Amazonia (CAAE: 19380913.6.3001.0005). Participants gave written consent after receiving information about the study objective and procedures.

### Study design and participants

This is a retrospective case series of dermatopathological findings using clinical, laboratory and histopathological data observed in 22 patients diagnosed with *Bothrops atrox* envenomation at FMT-HVD from August 2014 to August 2016. Patients were eligible if admitted to the hospital with less than 24 hours after the bite, without antivenom therapy in another hospital and without any sign of secondary infection at that time. On admission, *Bothrops atrox* snakebites were diagnosed based on clinical and epidemiological characteristics or, when the patient brought the snake responsible for the envenomation. Snake identification was performed by a zoologist from the FMT-HVD research group using a validated taxonomic key [27]. *B. atrox* is the only species of this genus in the study region [25,26]. Antivenom treatment was given to all patients according to clinical severity following the guidelines of the Brazilian Ministry of Health [28] and cases were classified as (1) mild: local pain, swelling, and bruising; (2) moderate: local manifestations without necrosis and minor systemic signs (coagulopathy and bleeding, no shock); or (3) severe: life-threatening snakebite, with severe bleeding, hypotension, shock, and/or acute kidney failure. Antivenom treatment was given 30 minutes after premedication with intravenous hydrocortisone (500 mg), intravenous cimetidine (300 mg), and oral dexchlorpheniramine (5 mg) (standardized according to local guidelines). Patients underwent local wound care, i.e., daily cleaning with 0.9% saline. Per the hospital protocol for pain, 1 g of intravenous metamizole every 6 hours was given on demand. Persistent intense pain was treated with 100 mg of intravenous tramadol.

 The affected limb was treated in the most comfortable position, according to patient preferences. When present, blisters were aspirated, necrotic tissue was surgically debrided, abscesses were drained, and antibiotic treatment was given accordingly.

### Epidemiological and laboratory data collection

After inclusion of the patient, demographic and epidemiological data were collected using a standardized form. Baseline data included information on sex, age (in years), area of occurrence (urban or rural), anatomical site of the bite, time elapsed from bite to medical assistance (in hours), and preadmission treatments (use of topical or oral medicines, use of tourniquet and other procedures).

Laboratory tests were performed according to FMT-HVD service routine and the Brazilian National Program of Quality Control (NPQC) and included the following variables and its reference values: leukocyte count (4.000 to 10.000/mm$^3$), platelet count (150.000 to 450.000/mm$^3$), hemoglobin (12,5 to 15,5 g/dl), creatine kinase (24 to 190 U/L), lactate dehydrogenase (211 to 423 U/L), Lee-White clotting time (5 to 10 minutes) and fibrinogen (180 to 350 mg/dL). All demographic, clinical, and laboratory information was collected through a standardized clinical registration form (REDCap, Vanderbilt University).

### Clinical evaluation

A clinical characterization was performed on admission and over the course of patient follow up. The presence of pain, fang marks, edema, erythema, ecchymosis, serum secretions, and perilesional bleeding was assessed. The extent of edema in the affected limb was also evaluated by measuring the distance between the distal and proximal points showing swelling. Edema was classified as mild (affecting 1–2 limb segments), moderate (affecting 3–4 segments) and severe (affecting more than 5 limb segments). Also, patients' clinical evolution regarding emerging of local complications such as necrosis, blisters, secondary infection (defined as the presence of cellulitis and/or abscess) and compartment syndrome were also collected. Any systemic signs and symptoms, such as systemic bleeding, acute kidney failure, headache, dizziness, and vomiting, were recorded.

### Skin sample collection

Skin biopsies were collected from the bite and local tissue damage area of all individuals included, but only after reversal of venom-induced consumption coagulopathy (where present) to mitigate the risk of bleeding and if the patient presented suggestive signs of local complications (e.g., necrosis, secondary infection). After local asepsis and antisepsis, lidocaine 2% was applied as a topical anesthesia. Then, a punch biopsy of 4 mm was performed. In patients who underwent surgical procedures such as debridement or placement of Penrose drain, a sample of tissue was collected at that moment by a dermatologist. Tissue samples were then immersed in formalin 10% for fixation.

### Histopathological data

Tissue slides were reviewed by two blinded pathologists (L.C.L.F and M.F.d.R) and a severity grading was applied based on the number of cells and other alterations that comprised the microscopic field, divided into 3 equal parts, at 200x magnification using 0500R microscope from OPTICOM. Severity grade of inflammatory infiltrate, fibrin thrombi, edema, congestion, and hemorrhage were classified as mild (1/3 of microscopic field comprised), moderate (2/3) and severe (3/3).

Sections were stained with hematoxylin-eosin and microscopical variables considered in evaluation, based on available dermatopathology textbook data, experimental and review literature data on snakebite envenomation local pathology [8,29,30]. These were as follows: changes in the epidermis (hyperkeratosis; acanthosis; exocytosis; spongiosis; blister formation;

thinning and detachment of the epidermis), as well as in the dermis and hypodermis (hemorrhage; inflammatory infiltrate—presence of histiocytes, lymphocytes, eosinophils, other granulocytes, or plasma cells—edema, congestion; collagen alterations; vascular damage; fibrin thrombi; necrosis, signs of infection and signs of tissue repair).

### Data analysis and presentation

Descriptive analysis was performed using *Stata (v13.0)*. Median and interquartile range was used in continuous data and percentile frequency distribution was used for categorical variables presented in text. Tables were used to present an individual description of all patients' demographic, clinical, laboratory and histopathological data. Figures were constructed based on patients who presented a variety of clinical and histopathological (macroscopical and microscopical) features.

## Results

### Participant characteristics

The study population consisted of 22 patients who underwent biopsy. **Table 1** summarizes the features of the study population. Males were predominant (77.3%), the mean age was 37 years (range, 18–63 years), and most participants lived in rural areas (90.9%). Lower limbs were the most affected region by the bite (72.3%). The average time from bite to medical care was 4 hours (*IQR* 3–5 hours) and 15 (68.2%) patients used a product and/or did a procedure on the bite site prior to arrival at medical care.

All patients presented pain and edema on admission. Edema was classified as moderate in 11 cases (50%), mild in 8 (36.4%) and severe in 3 (13.6%). Other frequent local findings were perilesional bleeding (72.3%) and ecchymosis (22.7%). Systemic alterations were observed in 13 patients (59.1%). Headaches were the most common (40.9%), followed by nausea (18.2%). Moreover, vomiting, sweating, and dizziness were present in 2 cases each (9.1%), and tachycardia and anuria, each observed in 1 case (4.5%). Seven cases (31.8%) presented systemic bleeding, with 4 cases (18.2%) of gum bleeding, and hematuria, hemoptysis, and hematochezia observed in 1 case each (4.5%). Most cases were classified as moderate (63.6%), followed by mild (22.7%) and severe (13.7%). Regarding laboratory findings, blood was incoagulable in 13 patients (59.1%), and fibrinogen was below the reference value in 14 (63.3%) cases. Creatine kinase and lactate dehydrogenase activities were above the reference value in 11 (50%) and 4 (18.2%) patients, respectively. Hemoglobin levels were below the reference value in 1 patient (4.5%) and values above the reference range of leukocyte counts were observed in 16 patients (72.3%). Additionally, 2 patients (9.1%) presented with thrombocytopenia and 1 (4.5%) with thrombocytosis.

### Clinical and laboratory findings on the day of tissue collection

**Table 2** presents the laboratory and clinical features of *Bothrops* snakebite patients on the day of tissue collection, as well as their histological alterations. On the time between snakebite and biopsy, although there was variation among patients, most procedures were performed during the first 4 days (90.9%) after the snakebite (6 patients on days 1 and 2, and 4 patients on days 3 and 4) (Figs 1, 2 and 3). One patient underwent biopsy on day 5 and another on day 8 (Figs 4 and 5). Most patients had no systemic alterations, but in 2 cases we observed acute kidney injury (50%) and diarrhea (50%). Laboratory findings showed normal clotting time (CT) in all patients. Moreover, 17 (77.2%) patients had fibrinogen levels evaluated on biopsy day, and 8/17 (47%) presented levels above the reference values, whereas 1 (5.88%) patient had lower

**Table 1. Epidemiological and clinical data of *Bothrops* envenomations on admission.**

| ID | Sex/ Age (years) | Snakebite classification | Antivenom vials | Time from bite to treatment (hours) | Self-care procedures | Bite site | Clinical and laboratory findings |
|---|---|---|---|---|---|---|---|
| 1 | M/52 | Mild | 4 | 4h | Garlic | Left foot | Local manifestations: Pain, mild edema, lymphadenomegaly and fang marks.<br>Systemic manifestations: Headache and nausea.<br>Laboratory tests: Incoagulable blood; Fibrinogen: undetectable; CK: 67 IU/L; LDH: 303 IU/L; Hemoglobin: 16.3 g/dl; Leukocytes: 15,020/mm$^3$; Platelets: 262,000/mm$^3$. |
| 2 | M/21 | Moderate | 8 | 3h | No | Left hand | Local manifestations: Pain, mild edema, bleeding, and fang marks.<br>No systemic manifestations.<br>Laboratory tests: Increased clotting time; Fibrinogen: 198 mg/dL; CK: 135 IU/L; LDH: 270 IU/L; Hemoglobin: 16.3 g/dl; Leukocytes: 12,070/mm$^3$; Platelets: 198,000/mm$^3$. |
| 3 | F/26 | Moderate | 8 | 3h | No | Left leg | Local manifestations: Pain, moderate edema, bleeding, and fang marks.<br>No systemic manifestations.<br>Laboratory tests: Normal clotting time; Fibrinogen: undetectable; CK: 161 IU/L; LDH: 318 IU/L; Hemoglobin: 13.5 g/dl; Leukocytes: 10,800/mm$^3$; Platelets: 225,000/mm$^3$. |
| 4 | M/19 | Moderate | 8 | 3h | Tourniquet and tucumã pulp (*Astrocaryum aculeatum*) | Right foot | Local manifestations: Pain, mild edema, bleeding, lymphadenomegaly and fang marks.<br>Systemic manifestations: Headache and gum bleeding.<br>Laboratory tests: Incoagulable blood; Fibrinogen: *na; CK: 416 IU/L; LDH: 395 IU/L; Hemoglobin: 14.9 g/dl; Leukocytes: 15,780/mm$^3$; Platelets: 215,000/mm$^3$. |
| 5 | F/18 | Moderate | 8 | 5h | Saline solution and tourniquet | Right ankle | Local manifestations: Pain, moderate edema, local bleeding, erythema, blood crust and blister.<br>No systemic manifestations.<br>Laboratory tests: Increased clotting time; Fibrinogen: 152 mg/dL; CK: 92 IU/L; LDH: 237 IU/L; Hemoglobin: 11.3 g/dl; Leukocytes: 15,170/mm$^3$; Platelets: 204,000/mm$^3$. |
| 6 | M/56 | Moderate | 8 | 2h | No | Right leg | Local manifestations: Pain, moderate edema, and fang marks.<br>No systemic manifestations.<br>Laboratory tests: Increased clotting time; Fibrinogen: 272 mg/dL; CK: 222 IU/L; LDH: 349 IU/L; Hemoglobin: 14.9 g/dl; Leukocytes: 8,010/mm$^3$; Platelets: 221,000/mm$^3$. |
| 7 | M/36 | Moderate | 8 | 5h | Alcohol | Both hands | Local manifestations: Pain, moderate edema, ecchymosis, and local bleeding.<br>Systemic manifestations: Tachycardia, sweating, vomiting.<br>Laboratory tests: Incoagulable blood; Fibrinogen: *na; CK: 98 IU/L; LDH: 337 IU/L; Hemoglobin: 15.69 g/dl; Leukocytes: 18,290/mm$^3$; Platelets: 188,000/mm$^3$. |
| 8 | M/63 | Mild | 4 | 3h | No | Left foot | Local manifestations: Pain, mild edema, bleeding, serum secretion and fang marks.<br>Systemic manifestations: Headache.<br>Laboratory tests: Incoagulable blood; Fibrinogen: undetectable; CK: 199 IU/L; LDH: 255 IU/L; Hemoglobin: 14.9 g/dl; Leukocytes: 7,360/mm$^3$; Platelets: 191,000/mm$^3$. |
| 9 | M/23 | Moderate | 8 | 8h | No | Right ankle | Local manifestations: Pain, moderate edema, bleeding, and fang marks.<br>Systemic manifestations: Gum bleeding.<br>Laboratory tests: Incoagulable blood; Fibrinogen: undetectable; CK: 491 IU/L; LDH: 383 IU/L; Hemoglobin: 16.06 g/dl; Leukocytes: 20,000/mm$^3$; Platelets: 90000/mm$^3$. |

(*Continued*)

**Table 1.** (*Continued*)

| ID | Sex/Age (years) | Snakebite classification | Antivenom vials | Time from bite to treatment (hours) | Self-care procedures | Bite site | Clinical and laboratory findings |
|---|---|---|---|---|---|---|---|
| 10 | F/60 | Moderate | 8 | 17h | Alcohol | Right ankle | Local manifestations: Pain, moderate edema, lymphadenomegaly, ecchymosis, and blister. Systemic manifestations: Headache, nausea, vomiting, sweating, hematuria, hematochezia, and acute kidney injury. Laboratory tests: Incoagulable blood; Fibrinogen: undetectable; CK: 232 IU/L; LDH: 911 IU/L; Hemoglobin: 13.7 g/dl; Leukocytes: 14,550/mm$^3$; Platelets: 180,000/mm$^3$. |
| 11 | M/27 | Mild | 8 | 2h | Alcohol and lemon juice | Left foot | Local manifestations: Pain, mild edema, bleeding, paresthesia, and serous secretion. No systemic manifestations. Laboratory tests: Normal clotting time; Fibrinogen: undetectable; CK: 122 IU/L; LDH: 303 IU/L; Hemoglobin: 14.7 g/dl; Leukocytes: 7,010/mm$^3$; Platelets: 227,000/mm$^3$. |
| 12 | M/22 | Severe | 10 | 1h | No | Right ankle | Local manifestations: Pain, severe edema, bleeding, lymphadenomegaly, erythema, ecchymosis, serous secretion, and fang marks. Systemic manifestations: Headache, nausea, hemoptysis. Laboratory tests: Incoagulable blood; Fibrinogen: 102 mg/dL; CK: 185 IU/L; LDH: 206 IU/L; Hemoglobin: 14.8 g/dl; Leukocytes: 15,870/mm$^3$; Platelets: 480,000/mm$^3$. |
| 13 | M/59 | Moderate | 5 | 4h | Gasoline | Left ankle | Local manifestations: Pain, moderate edema, and fang marks. No systemic manifestations. Laboratory tests; Increased clotting time; Fibrinogen: undetectable; CK: 72 IU/L; LDH: 310 IU/L; Hemoglobin: 14.0 g/dl; Leukocytes: 6,980/mm$^3$; Platelets: 201,000/mm$^3$. |
| 14 | M/46 | Moderate | 8 | 3h | Cooking oil | Left hand | Local manifestations: Pain, moderate edema, bleeding, and fang marks. Systemic manifestations: Headache, gum bleeding, and dizziness. Laboratory tests: Normal clotting time; Fibrinogen: 314 mg/dL; CK: 415 IU/L; LDH: 327 IU/L; Hemoglobin: 14.1 g/dl; Leukocytes: 5,120/mm$^3$; Platelets: 380,000/mm$^3$. |
| 15 | M/59 | Moderate | 8 | 6h | Coffee, lemon juice, and salt | Right leg | Local manifestations: Pain, severe edema, erythema, and bleeding. Systemic manifestations: gum bleeding and acute kidney injury. Laboratory tests: Normal clotting time; Fibrinogen: *na; CK: 399 IU/L; LDH: 347 IU/L; Hemoglobin: 14.9 g/dl; Leukocytes: 11,000/mm$^3$; Platelets: 136,000/mm$^3$. |
| 16 | M/36 | Severe | 12 | 6h | No | Left hand | Local manifestations: Pain, severe edema, ecchymosis, local bleeding, lymphadenomegaly, and fang marks. Systemic manifestations: acute kidney injury. Laboratory tests: Incoagulable blood; Fibrinogen: undetectable; CK: 645 IU/L; LDH: 560 IU/L; Hemoglobin: 15.7 g/dl; Leukocytes: 14,500/mm$^3$; Platelets: 226,000/mm$^3$. |
| 17 | M/21 | Moderate | 8 | 4h | Iodine | Left foot | Local manifestations: Pain, moderate edema, erythema, local bleeding, and fang marks. Systemic manifestations: Headache and nausea. Laboratory tests: Normal clotting time; Fibrinogen: *na; CK: 217 IU/L; LDH: 383 IU/L; Hemoglobin: 16.0 g/dl; Leukocytes: 16,620/mm$^3$; Platelets: 268,000/mm$^3$. |
| 18 | M/38 | Mild | 8 | 3h | Cooking oil | Left foot | Local manifestations: Pain, mild edema, bleeding, and bite marks. Systemic manifestations: Headache. Laboratory tests: Incoagulable blood; Fibrinogen: *na; CK: 308 IU/L; LDH: 469 IU/L; Hemoglobin: 15.6 g/dl; Leukocytes: 5,510/mm$^3$; Platelets: 223,000/mm$^3$. |

(*Continued*)

**Table 1.** (*Continued*)

| ID | Sex/ Age (years) | Snakebite classification | Antivenom vials | Time from bite to treatment (hours) | Self-care procedures | Bite site | Clinical and laboratory findings |
|---|---|---|---|---|---|---|---|
| 19 | F/57 | Moderate | 8 | 20h | Cooking oil, onion and tourniquet | Right hand | Local manifestations: Pain, moderate edema, and fang marks No systemic manifestations. Laboratory tests: Incoagulable blood; Fibrinogen: 61 mg/dL; CK: 64 IU/L; LDH: 336 IU/L; Hemoglobin: 14.8 g/dl; Leukocytes: 10,080/mm$^3$; Platelets: 257,000/mm$^3$. |
| 20 | M/20 | Mild | 8 | 5h | Pirarucu's fat (*Arapaima gigas*) | Left leg | Local manifestations: Pain, mild edema, bleeding, and fang marks. Systemic manifestations: Dizziness. Laboratory tests: Incoagulable blood; Fibrinogen: undetectable; CK: 180 IU/L; LDH: 297 IU/L; Hemoglobin: 15.3 g/dl; Leukocytes: 11,670/mm$^3$; Platelets: 299,000/mm$^3$. |
| 21 | M/28 | Moderate | 8 | 4h | Tourniquet | Left foot | Local manifestations: Pain, mild edema, and erythema. Systemic manifestations: Headache. Laboratory tests: Incoagulable blood; Fibrinogen: undetectable; CK: 289 IU/L; LDH: 500 IU/L; Hemoglobin: 14.2 g/dl; Leukocytes: 14,530/mm$^3$; Platelets: 156,000/mm$^3$. |
| 22 | F/29 | Severe | 12 | 6h | Balsam (infusion of different plants in vegetal oil) | Right hand | Local manifestations: Pain, moderate edema, ecchymosis, erythema, and blister. No systemic manifestations. Laboratory tests: Incoagulable blood; Fibrinogen: undetectable; CK: 351 IU/L; LDH: 368 IU/L; Hemoglobin: 13.8 g/dl; Leukocytes: 30,530/mm$^3$; Platelets: 384,000/mm$^3$. |

M: Male; F: Female. No = not observed in patients. *na = data not available. CT = clotting time. CK = creatine phosphokinase. LDH = lactate dehydrogenase. Reference values: Lee-White clotting time ≤10 min; Fibrinogen: 180–350 mg/dL; Creatine kinase: 24–190 IU/L; Lactate dehydrogenase: 211–423 IU/L; Hemoglobin: 12.5–15.5 g/dl; Leukocytes: 4,000–10,000/mm$^3$; Platelets: 150,000–450,000/mm$^3$.

values. Creatine kinase and lactate dehydrogenase were above normal range levels in 9/22 (40.9%) and 4/21 (19%) patients, respectively. Leukocytosis was observed in 7 (21.8%), and thrombocytopenia in 2 (9%). Also, hemoglobin levels were increased in 3 patients (13.6%) and below reference values in 6 (27.2%) cases. Regarding clinical local features, edema was observed in all patients, varying from mild (54.5%), to moderate (27.7%) and severe (18.1%). Other common findings were fang marks (63.6%), secondary infection (36.3%), ecchymosis (31.8%) and erythema (22.7%), while blistering and necrosis were also present in a small number of patients, specifically 3 (13.6%) and 1 (4.5%), respectively (Figs 1A–5A).

### Histopathological findings in *Bothrops* snakebite patients

On histopathological evaluation, out of 22 patients only 9 (40.9%) showed epidermal alterations. Spongiosis was observed in 3 cases (33.3%), with hyperkeratosis and acanthosis in 2 (22.2%) cases each (Figs 2, 4 and 5). Also, hyperplasia was found in 1 (11.1%) patient, as well as hemorrhagic intraepidermal blister with neutrophil exocytosis (11.1%) (Fig 2C) and severe epidermis necrosis (11.1%). One case had no epidermis but was attributed to technical artifacts rather than a consequence of snakebite envenomation. Changes in dermis and hypodermis consisted mainly of hemorrhage, inflammatory infiltrate, vascular injury, edema, and congestion, although there were cases of collagen damage, necrosis, abscess, and presence of granulation tissue signalizing repair (Table 2).

Hemorrhage was found in 15 out of 22 (68.1%) patients and could be observed as mild (46.6%), moderate (20%) or severe (33.4%). Edema and congestion were described in 18 cases (81.8%), classified as mild (44.4%), moderate (44.4%) and severe (11.2%). Moreover,

**Table 2. Clinical data of *Bothrops* envenomation patients on biopsy day.**

| ID | Time from bite to biopsy (hours) | Systemic findings | Laboratory findings | Biopsy site | Local findings | Hystopathological findings |
|---|---|---|---|---|---|---|
| 1 | 24 | No | Normal clotting time and fibrinogen levels, CK: 47 IU/L, LDH: 245 IU/L, Hemoglobin:16.4 g/dl, Leukocytes: 10,030 /mm³, Platelets: 224,000/mm³. | Between 3rd and 4th left toes | Mild edema | Severe hemorrhage, mild inflammatory infiltrate (granulocytes, neutrophils), mild edema and congestion, and mild vascular injury on hypodermis |
| 2 | 24 | No | Normal clotting time and fibrinogen levels, CK: 218 IU/L, LDH: 549 IU/L, Hemoglobin: 15.6 g/dl, Leukocytes: 16,440/ mm³, Platelets: 190,000/mm³. | 5th digit of left hand | Mild edema | Spongiosis; mild hemorrhage, moderate edema and congestion, moderate mixed inflammatory infiltrate (granulocytes, neutrophils, histiocytes and lymphocytes), moderate vascular injury and fibrin thrombi on deep dermis and hypodermis; moderate signs of tissue repair. |
| 3 | 24 | No | Normal clotting time and fibrinogen levels, CK: 87 IU/L, LDH: 224 IU/L, Hemoglobin: 13.2 g/dl, Leukocytes: 12,000/mm³, Platelets: 211,000/mm³. | Distal third of lower left leg | Moderate edema | Severe hemorrhage on deep dermis, mild congestion and edema, moderate inflammatory infiltrate (granulocytes, neutrophils) and mild vascular injury on hypodermis. |
| 4 | 24 | No | Normal clotting time and fibrinogen levels, CK: 390 IU/L, LDH: 528 IU/L, Hemoglobin: 16.3 g/dl, Leukocytes: 15,510/ mm³, Platelets: 133,000/mm³. | Right foot | Mild edema | Severe inflammatory infiltrate of granulocytes and neutrophils, moderate hemorrhage, mild edema and congestion, and moderate collagen fibers necrosis |
| 5 | 24 | No | Normal clotting time and high fibrinogen levels (376 mg/dL), CK: 83 IU/L, LDH: 419 IU/L, Hemoglobin: 13.1 g/dl, Leukocytes: 13,140/mm³, Platelets: 194,000/mm³. | Right ankle | Moderate edema | Mild inflammatory infiltrate (histiocytes and lymphocytes) on dermis. |
| 6 | 24 | No | Normal clotting time and fibrinogen levels, CK: 234 IU/L, LDH: 223 IU/L, Hemoglobin: 13.8 g/dl, Leukocytes: 6,840/ mm³, Platelets: 163,000/mm³. | Right calf | Moderate edema and secondary infection (abscess and cellulitis) | Severe hemorrhage and mild fibrin thrombi formation on deep dermis and hypodermis. |
| 7 | 48 | No | Normal clotting time and fibrinogen levels, CK: 57 IU/L, LDH: 212 IU/L, Hemoglobin: 12.9 g/dl, Leukocytes 9,050/mm³, Platelets: 143,000/mm³. | 2nd digit of both hands | Left: edema and ecchymosis. Right: moderate edema, erythema, ecchymosis, and blister with blood content | Left finger: normal skin without any alterations. Right finger: moderate edema and congestion, moderate vascular injury, and moderate fibrin thrombi formation; severe inflammatory infiltrate of granulocytes and neutrophils and mild presence of histiocytes and lymphocytes. |
| 8 | 48 | No | Normal clotting time and fibrinogen levels, CK: 346 IU/L, LDH: 187 IU/L, Hemoglobin: 13.8 g/dl, Leukocytes: 7,980/ mm³, Platelets: 173,000/mm³. | 2nd toe of left foot | Mild edema and ecchymosis | Severe epidermis' necrosis; mild edema and congestion, mild mixed inflammatory infiltrate (granulocytes, neutrophils, histiocytes and lymphocytes), mild vascular injury and mild thrombi formation on dermis; mild tissue repair process can also be observed. |
| 9 | 48 | No | Normal clotting time and high fibrinogen levels (487 mg/dL), CK: 1187 IU/L, LDH: 383 IU/L, Hemoglobin: 13.5 g/dl, Leukocytes: 14,110/mm³, Platelets: 147,000/mm³. | Medial malleolus of right foot | Moderate edema, ecchymosis, and secondary infection (abscess) | Dermis and hypodermis showing severe hemorrhage, moderate edema, and congestion; severe inflammatory infiltrate of granulocytes and neutrophils and mild presence of histiocytes and lymphocytes; moderate vascular injury and mild fibrin thrombi formation. |

(*Continued*)

**Table 2.** (*Continued*)

| ID | Time from bite to biopsy (hours) | Systemic findings | Laboratory findings | Biopsy site | Local findings | Hystopathological findings |
|---|---|---|---|---|---|---|
| 10 | 48 | Acute kidney injury | Normal clotting and high fibrinogen levels (416 mg/dL), CK: 111 IU/L, LDH: 616 IU/L, Hemoglobin: 12.4 g/dl, Leukocytes: 8160/mm$^3$, Platelets: 15,1000/mm$^3$. | Right ankle | Mild edema and ecchymosis | Severe spongiosis and hemorrhagic blister, and mild exocytosis on epidermis; mild hemorrhage, severe edema and congestion, severe inflammatory infiltrate of granulocytes and neutrophils and mild presence of histiocytes and lymphocytes, severe vascular injury, and severe fibrin thrombi on dermis. |
| 11 | 48 | No | Normal clotting time and fibrinogen levels, CK: 71 IU/L, LDH: 240 IU/L, Hemoglobin: 15.1 g/dl, Leukocytes: 6,650/mm$^3$, Platelets: 196,000/mm$^3$. | Left foot near the 1st toe | Mild edema and ecchymosis | Mild hemorrhage, mild edema and congestion, moderate inflammatory infiltrate (granulocytes and neutrophils) and moderate collagen hyaline necrosis. |
| 12 | 48 | No | Normal clotting time and fibrinogen levels, CK: 123 IU/L, LDH: 184 IU/L, Hemoglobin: 14.2 g/dl, Leukocytes 7,400/mm$^3$, Platelets: 96,000/mm$^3$. | Distal third of lower left leg | Severe edema, erythema, and secondary infection (cellulitis) | Severe hemorrhage, severe inflammatory infiltrate (granulocytes, neutrophils), moderate edema and congestion, mild vascular injury, and mild thrombi formation; mild collagen fibers lysis, hyaline and skin appendage necrosis. |
| 13 | 72 | No | Normal clotting and high fibrinogen levels (457 mg/dL),CK: 47 IU/L, LDH: 263 IU/L, Hemoglobin: 14.4 g/dl, Leukocytes: 8,180/mm$^3$, Platelets: 192,000/mm$^3$. | Medial malleolus of left foot | Mild edema and fang marks | Mild hemorrhage, moderate edema and congestion, mild vascular injury and severe inflammatory infiltrate of granulocytes and neutrophils, mild presence of histiocytes and lymphocytes, and mild vascular injury on dermis. |
| 14 | 72 | No | Normal clotting time and fibrinogen levels, CK: 279 IU/L, LDH: 265 IU/L, Hemoglobin: 14.8 g/dl, Leukocytes: 5,760/mm$^3$, Platelets: 301,000/mm$^3$. | Dorsum of left hand | Moderate edema and fang marks | Hyperkeratosis; mild inflammatory infiltrate (histiocytes and lymphocytes) on dermis. |
| 15 | 72 | Acute kidney injury | Normal clotting time and fibrinogen levels, CK: 713 IU/L, LDH: 257 IU/L, Hemoglobin: 11.9 g/dl, Leukocytes: 6,900/mm$^3$, Platelets: 111,000/mm$^3$. | Middle third of lower right leg | Severe edema, secondary infection (abscess and cellulitis) and compartment syndrome | Mild edema and congestion, mild inflammatory infiltrate of granulocytes and neutrophils and mild vascular injury on dermis. |
| 16 | 72 | Acute kidney injury | Normal clotting time and high fibrinogen levels (446 mg/dL), CK: 200 IU/L, LDH: 488 IU/L, Hemoglobin: 10.9 g/dl, Leukocytes: 8,220/mm$^3$, Platelets: 151,000/mm$^3$. | Inner surface of left hand | Severe edema, ecchymosis, erythema, and secondary infection (cellulitis) | Hyperkeratosis and acanthosis; on dermis mild inflammatory infiltrate of histiocytes and lymphocytes. |
| 17 | 96 | No | Normal clotting time and fibrinogen levels, CK: 163 IU/L, Hemoglobin: 14.4 g/dl, Leukocytes: 6,390/mm$^3$, Platelets: 214,000/mm$^3$. | 2nd toe of left foot | Mild edema, erythema, ecchymosis, blister, and secondary infection | Hyperplasia on epidermis; moderate hemorrhage, moderate edema and congestion, severe inflammatory infiltrate (granulocytes and neutrophils) on dermis; severe necrosis, severe eosinophilic infiltrate, and moderate abscess present on hypodermis. |
| 18 | 96 | No | Normal clotting time and fibrinogen levels, CK: 107 IU/L, LDH: 358 IU/L, Hemoglobin: 15.3 g/dl, Leukocytes: 6,180/mm$^3$, Platelets: 192,000/mm$^3$. | 4th toe of left foot | Mild edema and fang marks | No epidermis; mild hemorrhage, severe edema and congestion, and severe inflammatory infiltrate of neutrophils and eosinophils on dermis. |
| 19 | 96 | No | Normal clotting time and fibrinogen levels, CK: 44 IU/L, LDH: 274 IU/L, Hemoglobin: 11.8 g/dl, Leukocytes 8,660/mm$^3$, Platelets: 225,000/mm$^3$. | 1st digit of right hand | Mild edema and fang marks | Mild edema and congestion, and mild mixed inflammatory infiltrate of granulocytes, neutrophils, histiocytes and lymphocytes on dermis. |

(*Continued*)

**Table 2.** (Continued)

| ID | Time from bite to biopsy (hours) | Systemic findings | Laboratory findings | Biopsy site | Local findings | Hystopathological findings |
|---|---|---|---|---|---|---|
| 20 | 96 | No | Normal clotting and high fibrinogen levels (602 mg/dL), CK: 126 IU/L, LDH: 185 IU/L, Hemoglobin: 13.8 g/dl, Leukocytes 8,360/mm$^3$, Platelets: 218,000/mm$^3$. | Left calf | Mild edema and fang marks | Moderate hemorrhage, moderate edema and congestion, moderate inflammatory infiltrate of granulocytes, neutrophils and mild presence of histiocytes and lymphocytes, mild vascular injury and moderate necrosis in hypodermis. |
| 21 | 120 | No | Normal clotting time and high fibrinogen levels (613 mg/dL), CK: 114 IU/L, LDH: 383 IU/L, Hemoglobin: 13.0 g/dl, Leukocytes: 9760/mm$^3$, Platelets: 247,000/mm$^3$. | Dorsum of left foot | Mild edema, erythema, and secondary infection (cellulitis) | Spongiosis; mild hemorrhage, mild edema and congestion, severe inflammatory infiltrate of granulocytes, neutrophils, and mild presence of histiocytes and lymphocytes, moderate vascular injury, and thrombi formation. Severe abscess present on hypodermis. |
| 22 | 192 | No | Normal clotting and high fibrinogen levels (637 mg/dL), CK: 89 IU/L, LDH: 393 IU/L, Hemoglobin: 10.1 g/dl, Leukocytes 12,550/mm$^3$, Platelets: 288,000/mm$^3$. | Lateral surface of right hand | Severe edema, erythema, necrosis, blister, and secondary infection (abscess) | Irregular acanthosis; mild hemorrhage, moderate edema and congestion, and moderate mixed inflammatory infiltrate (granulocytes, neutrophils, histiocytes and lymphocytes). The presence of granulation tissue indicates intense tissue repair. |

M: Male; F: Female. No = not observed in patients. *na = data not available. CT = clotting time. CK = creatine phosphokinase. LDH = lactate dehydrogenase.

Reference values: Lee-White clotting time ≤10 min; Fibrinogen: 180–350 mg/dL; Creatine kinase: 24–190 IU/L; Lactate dehydrogenase: 211–423 IU/L; Hemoglobin: 12.5–15.5 g/dl; Leukocytes: 4,000–10,000/mm$^3$; Platelets: 150,000–450,000/mm

inflammatory infiltrate (II) was present in 21 (95.4%) patients as mild (33.4%), moderate (23.8%) and severe (42.8%) (Figs 1B and 4D). Regarding the type of II, in 8 (38%) cases granulocytes and neutrophils were the most common cell types, with 2 (9.5%) cases of eosinophilic infiltration, whereas in 3 (14.2%) patients, histiocytes and lymphocytes were the main character. Also, 10 (47.6%) cases revealed a mixed type of inflammatory infiltration (Fig 5B3) (Table 2).

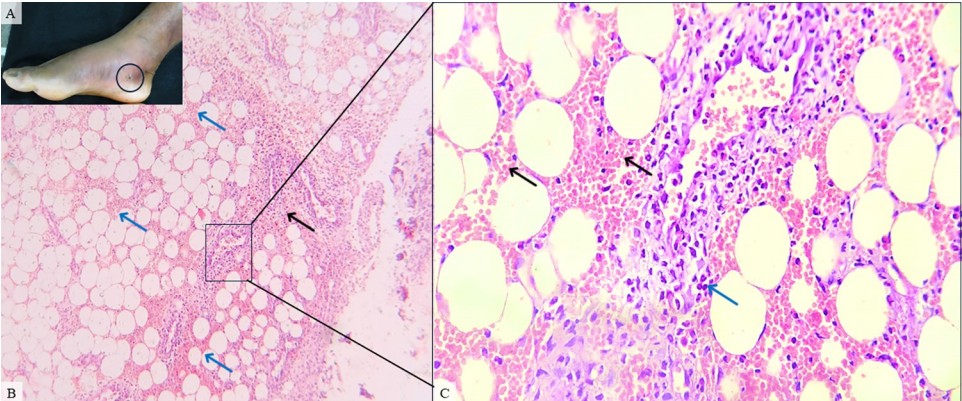

**Fig 1. Patient #9, male, 23, classified as a moderate case. The time elapsed between snakebite and antivenom treatment was 8 hours. A.** Lower right limb presenting moderate edema, ecchymosis, and biopsy suture (black circle) on day 4 after envenomation. **B.** Skin biopsy showing severe inflammatory infiltrate (black arrow), hemorrhage (blue arrow) and moderate vascular damage (inset) on hypodermis (HE, 100X). **C.** A magnified view of the erythrocytes (black arrow) and vascular damage (blue arrow) is shown in the inset (HE, 400X).

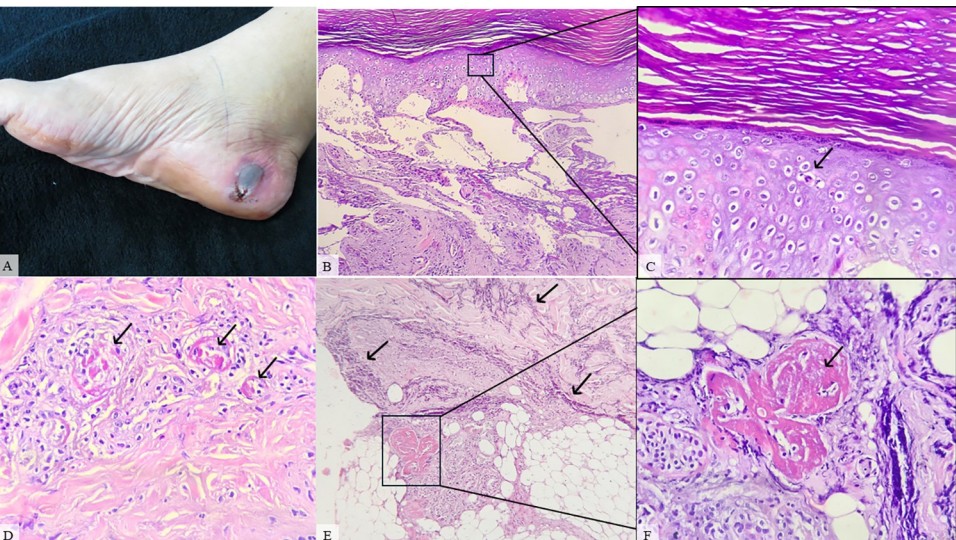

**Fig 2. Patient #10, female, 60, classified as a moderate case. Alcohol was used as a self-care procedure at the bite site and the time elapsed between snakebite and antivenom treatment was 17h30min. A.** Lower right limb presenting hemorrhagic blister (black circle) on day 4 after envenomation. **B.** Skin biopsy showing severe spongiosis that evolved to hemorrhagic blister and neutrophil exocytosis (HE, 100X). **C.** A magnified view of the neutrophil exocytosis (black arrow) is shown in the inset (HE, 400X). **D.** Vascular damage (black arrow) in the dermis (HE, 400x). **E.** Dermis and hypodermis presenting severe inflammatory infiltrate (black arrow) (HE, 100x). **F.** Inset showing a magnified view of fibrin thrombi (black arrow) in the deep dermis (HE, 400x).

In 12 (54.5%) patients some variety of vascular injury, e.g vasculitis, thickening of wall blood vessel, and/or microvascular damage, was observed (Figs 1–4), and formation of fibrin thrombi in vessel lumen was also present in 8 cases (36.3%) (Figs 2D and 4D), classified as mild (50%), moderate (37.5%) and severe (12.5%). Furthermore, other histopathological findings of the collected biopsies included collagen alterations (13.6%), in addition to necrosis (18.1%) (Fig 3B), abscess on the hypodermis (9%) (Fig 4C), and signs of tissue repair (13.6%) (Fig 5B4) (Table 2).

The histopathological findings described above could be observed separately or simultaneously in patients. For example, patient #12, a severe case of envenomation whose biopsy was performed 2 days after snakebite, presented with hemorrhage, inflammatory infiltrate, edema, congestion, vascular injury with thrombi formation and collagen damage such as fibers lysis, hyaline necrosis, and skin appendage necrosis. Likewise, patient #22, also a severe case but with biopsy performed 8 days after envenomation, showed irregular acanthosis, hemorrhage, edema, congestion, mixed inflammatory infiltration and granulation tissue, this last feature probably related to the later time in which the biopsy procedure was performed (Table 2 and Fig 6).

The histopathological features described in this study were non-specific in relation to snakebite classification or the time elapsed between envenomation and biopsy. Interestingly, persistence of distinctive histological patterns could be observed in almost all patients. Hemorrhage and inflammatory infiltrate were among the most common alterations and were found even 8 days after snakebite and antivenom administration, as well as dermal edema and congestion. Vascular damage and fibrin thrombi were also present in 12 and 8 patients, respectively, on different days of evaluation and degrees of severity. Furthermore, necrosis was observed in a small number of patients, as well as abscess formation, established as a sign of infection in dermis or hypodermis. Lastly, granulation tissue was observed in patient #22 at

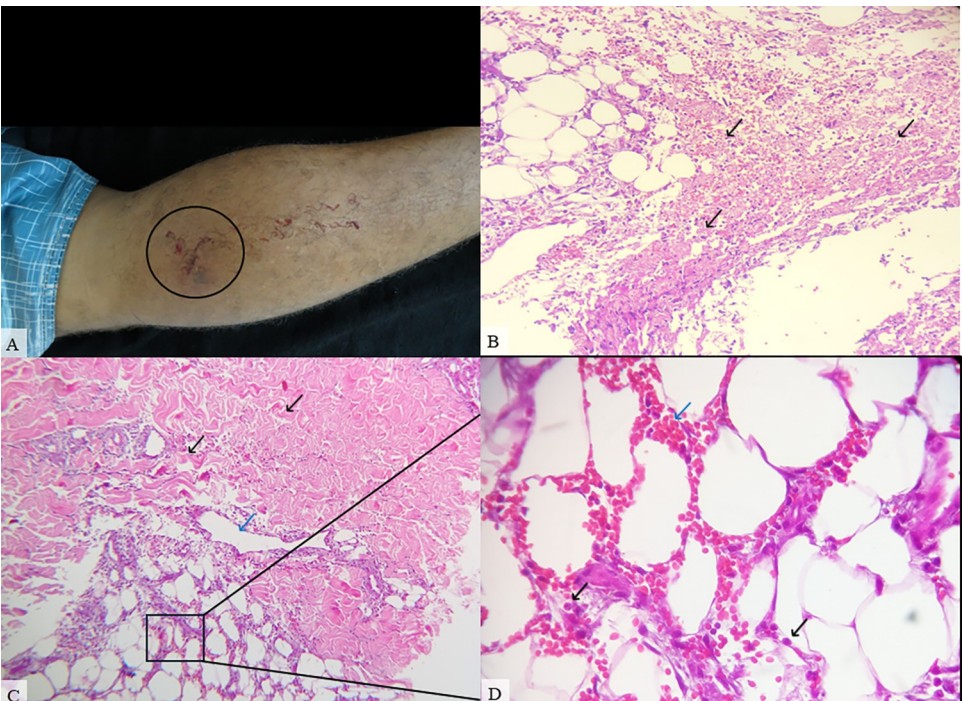

**Fig 3. Patient #20, male, 20, classified as a mild case. Pirarucu's fat (*Arapaima gigas*) was used as a self-care procedure at the bite site and the time elapsed between snakebite and antivenom treatment was 5 hours. A.** Lower left limb presenting mild edema and bleeding (black circle) on hospital admission. **B.** Hypodermis presenting severe necrosis (black arrow) (HE, 200X). **C.** Skin biopsy showing moderate dermal edema (black arrow), vascular damage (blue arrow), moderate hemorrhage and inflammatory infiltrate on hypodermis (HE, 100X). **D.** A magnified view of the inflammatory cells (black arrow) and erythrocytes (blue arrow) is shown in the inset (HE, 400X).

day 8 (196 hours) after envenomation, which was expected as the processes of tissue repair are already in place. However, in patients #2 and #8, whose biopsy took place within 2 days after snakebite, signs of tissue repair could also be observed; a circumstance that requires further investigation (**Fig 6**).

The stratification of the 22 patients included in this study into subgroups for comparison did not allow us to observe any significant difference in histopathological findings according to sex, age, severity classification, time to medical assistance, time from bite to biopsy, first aid used, clinical manifestations, and CK and LDH increase.

## Discussion

Tissue damage is an important feature in *Bothrops* envenomation, especially when it comes to human envenomation. Manifestations such as pain, edema, erythema, bleeding, and ecchymosis are relatively common and the emerging of complications like secondary infection, blisters, necrosis and compartment syndrome can lead to important consequences to the victims [12,31–33]. Here we presented a case series of dermatopathological findings in patients of *Bothrops* snakebites that occurred in the Brazilian Amazon.

Local and systemic manifestations in *Bothrops* envenomation may result from cytotoxic activities of venom toxins or the inflammatory process in response to tissue damage [34]. Direct action of venom includes degradation of extracellular matrix and/or components of the basement membrane, and hemostatic effects that contribute to bleeding disorders, vascular damage, and tissue destruction. Such effects also induce the production and activation of

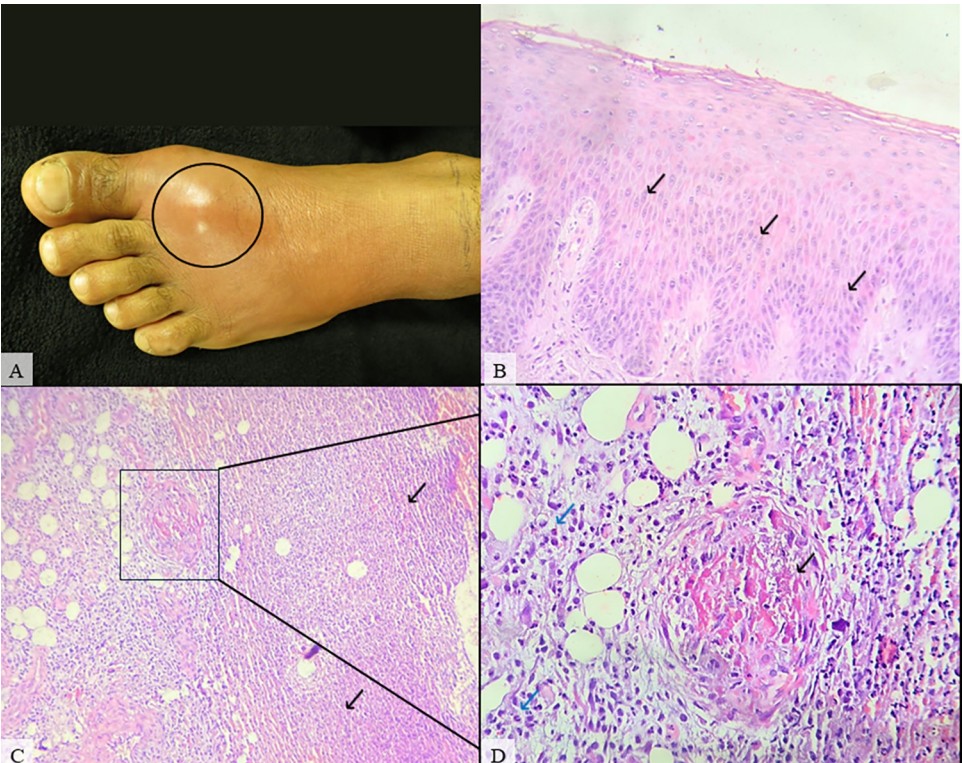

**Fig 4. Patient #21, male, 28, classified as a severe case. A tourniquet was applied as a self-care procedure near the bite site and the time elapsed between snakebite and antivenom treatment was 4 hours. A.** Lower left limb presenting cellulitis (black circle) on day 6 after envenomation. **B.** Skin biopsy showing moderate spongiosis (black arrow) (HE, 200X). **C.** Deep dermis presenting severe abscess (black circle), vascular damage and fibrin thrombi (HE, 100X). **D.** Vascular damage (black arrow) and severe inflammatory infiltrate (blue arrow) in the deep dermis (HE, 400x).

endogenous mediators that aim to act in regenerative process but can also enhance harmful tissue alterations [9,15–17,35,36]. These actions contribute to the pathophysiological effects observed in envenomation and changes in laboratory parameters presented by patients, which reflect both its systemic dynamics and extensive local damage promoted by the venom.

As a consequence to hemostatic effects caused by *Bothrops* venom, altered clotting time and low, or undetectable, fibrinogen levels are common features observed in patients on the first hours after envenomation and tend to normalize by 24 hours after antivenom administration [37], a similar situation observed in our study. Although, on biopsy day, fibrinogen levels presented higher levels than normal range. Fibrinogen is an acute phase protein, and its serum concentration may be increased in inflammatory and infectious conditions that are associated to vascular damage [38]. Furthermore, leukocytosis is also normally observed and is possibly related to the onset of inflammation caused by snakebite envenomation [39]. Changes in values of creatine kinase (CK) and lactate dehydrogenase (LDH) are relatively common findings in *Bothrops* envenomation. In our study, alterations in these parameters were observed both at admission and biopsy day. Previous studies carried out in Amazonas State reported high levels of CK and LDH in the first moments after envenomation, and LDH may reflect tissue damage in patients evaluated at extended time intervals [12,40,41].

Results reported in our case series regarding local manifestations are similar to other clinical studies of *Bothrops* envenomation conducted in Acre and Amazonas States, in which pain

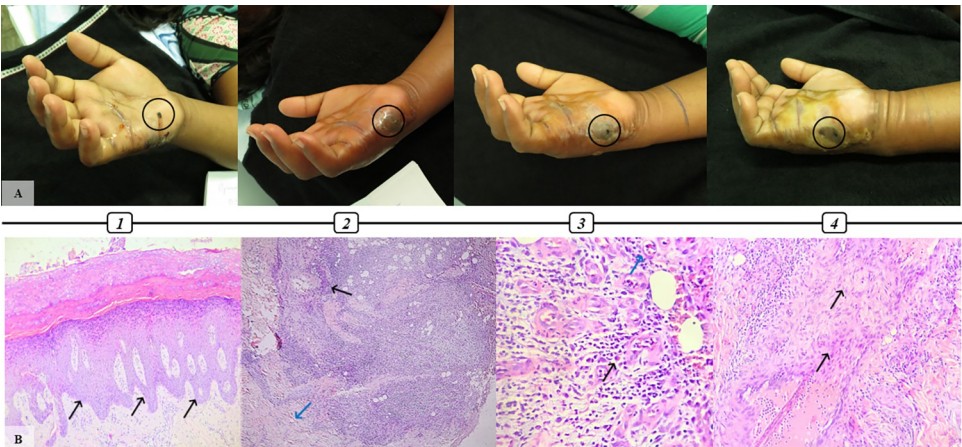

**Fig 5. Patient #22, female, 29, classified as a severe case. An infusion of different plants in vegetal oil, known as "balsam", was made as a self-care procedure at the bite site and the time elapsed between snakebite and antivenom treatment was 6 hours. Tissue damage evolution and microscopical alterations observed after envenomation. A1.** 2 days after snakebite presenting severe edema (from hand to upper arm) snakebite marks (black circle), serum secretion and blister formation. **A2.** Day 3 with severe edema, blisters on dorsal region spreading to hand palm and necrotic area (black circle). **A3.** Day 4 with edema, blisters, and necrosis on snakebite side (black circle). **A4.** Day 6 with edema, blisters with pus, necrosis on the snakebite site (black circle). A debridement was made to remove necrotic area on day 8. Tissue samples for histological analysis were obtained at that moment. **B1.** Biopsy presenting severe irregular acanthosis (black arrow) (HE, 100x). **B2.** Mixed inflammatory infiltrate (black arrow) and formation of new capillaries (blue arrow) responsible for collagen production in tissue repair process (HE, 100x). **B3.** Mixed inflammatory infiltrate (black arrow) and formation of new capillaries (blue arrow) (HE, 200x) **B4.** Activated fibroblasts (black arrow) (HE, 200x).

and edema were present in 40% of cases [12]. Blisters were observed in 10.2% of patients, as well as necrosis in 2.5%, and high frequence of secondary infection, reported in 40% of envenomation cases [12,42–44]. Local damage is a distinctive characteristic of *Bothrops* snakebites,

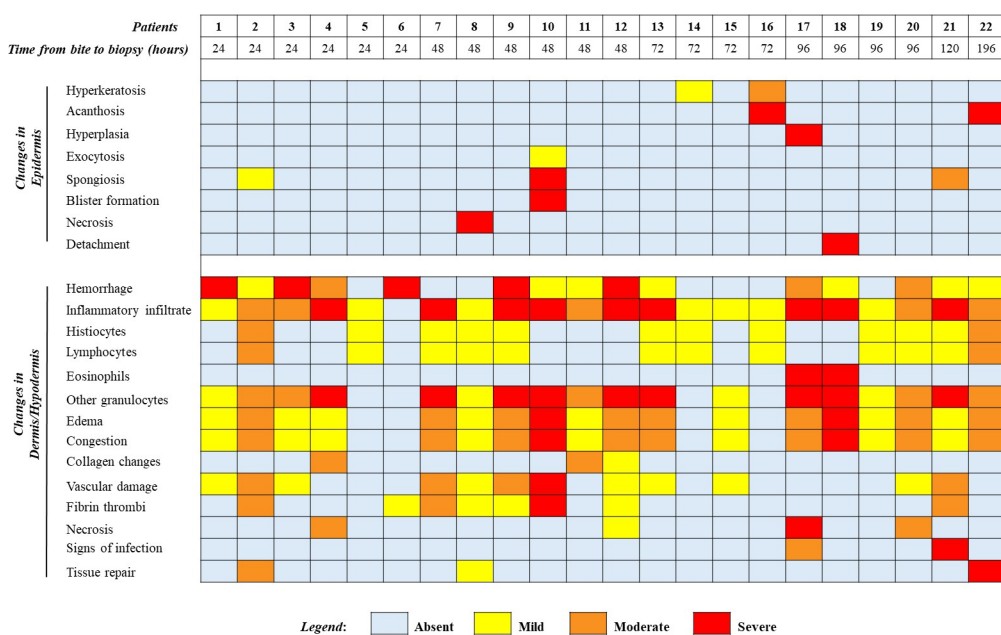

**Fig 6. Histopathological findings of *Bothrops* envenomation patients according to severity grade: absent, mild, moderate, and severe.** Patients were distributed in ascending order of time (in hours) from bite to the day of biopsy procedure.

with acute signs, like pain and edema, making an early appearance and others, such as blister formation, emerging well after envenomation. Nevertheless, all manifestations represent important findings and its evolution should be evaluated at all moments to establish patients conditions and recovery [45,46].

Regarding histological findings, epidermal alterations such as spongiosis, acanthosis and hyperkeratosis were the most common signs observed in our cases series, with isolated cases of hyperplasia, hemorrhagic intraepidermal blister and severe necrosis. These findings are similar to those reported in animal models and other types of envenomation [47,48]. As regarding humans, a case report on a fatal envenomation by *Naja kaouthia* have shown postmortem skin histopathology. The patient presented puncture wound in the forearm, that reached the epidermis layer, with necrosis of the epidermis and necrosis of the underlying dermis. Microscopic examination revealed areas of necrosis and acute inflammation in the skin and subcutaneous tissue sections. Acute vascular congestion was apparent in multiple separate tissue sites, with focal necrosis in vessels [49].

On hyperkeratosis and hyperplasia, this finding may be related to both envenomation, biopsy site and patients' habits, since many of them work in the field and the friction generated by this type of activity, as well as exposure to chemical aggressive compounds, can lead to an increase in the stratum corneum and/or other nonkeratinized skin layers [50]. *In vitro* and animal model studies have already reported the cytotoxic effect of L-amino acid oxidase (LAAOs), which is present in *Bothrops* venom, on human keratinocytes [51] and the destruction of the ECM that forms the dermo-epidermal junction by SVMPs [22,52]. Therefore, the presence of acanthosis may be related to the hyperproliferation of keratinocytes in response to envenomation [53]. Spongiosis probably occurred due to the increase in vascular permeability promoted by the release of inflammatory mediators after ECM destruction, while formation of subepidermal blisters is likely mainly due to the degradation of basement membrane (BM) components that connect epidermis to dermis, as already reported in human and experimental studies using *Bothrops* venom, and other types of envenomation [8,22,54–56].

Regarding our observed changes in the dermis and hypodermis, erythrocyte extravasation and influx of leukocytes, causing infiltration in the tissue, are commonly reported in animal model studies as well as human envenomation [13,30,53,57]. These changes occur from the hydrolysis of BM components of microvasculature which, combined with normal biophysical forces, lead to blood capillary destruction. Furthermore, the inflammatory response that results from envenomation contributes to the accumulation of leukocytes in situ, and production of bioactive molecules, such as cytokines, chemokines, growth factors and reactive oxygen species (ROS), that play an important part in local deleterious effects. Also, local hypoxia due to capillary obstruction or destruction may also be responsible for such alterations [13,15,57–61].

Collagen modifications, which were also reported in our study, may result from direct action of the venom as well as the action of endogenous proteinases, mainly matrix metalloproteinases (MMPs), which participate in physiological processes, but can have increased activity in response to inflammation following envenomation. Degradation of fibrillar collagens, mainly collagen types I and III, that compose connective tissue are most likely to happen by the latter process [13,62]. Furthermore, other alterations such as vascular damage, may originate from venom hemorrhagins (i.e. SVMPs) disrupting microvascular structure and stimulating acute inflammatory responses, leading to leukocytes infiltration *in situ*. Also, formation of fibrin thrombi was reported, and probably appears due to a prothrombotic state induced by hypercoagulability related to venom activity. In *Bothrops* envenomation, bleeding and thrombosis can appear simultaneously, a characteristic that was also observed in our patients [6,10,57,63,64].

Finally, formation of granulation tissue, an important component of the wound healing process, was also observed. Generally, tissue repair can occur with effective epithelial regeneration and revascularization resulting in restored skin integrity. However, in some cases, a deficient process may be present leading to fibrosis and scarring, as well as loss of function or nonhealing chronic wounds, e.g. as seen in diabetic foot ulcers. A study carried out in Sri Lanka with patients presenting with chronic wounds more than one month old after envenomation by the vipers *Daboia russelii* and *Hypnale* spp. and the cobra *Naja naja* concluded that granulation tissue and vascular proliferations were more present in snakebite wounds, when comparing them to other wounds of different etiology [65–67]. Although our work did not report patients with any chronic wounds, these literature findings reinforce the role of snakebite envenomation as a cause of disability and highlight the need for easy and rapid access to antivenom therapy, along with better clinical management of local tissue damage.

Currently the only treatment for snakebite envenomation is antivenom therapy, which effectively neutralizes systemic alteration [68]. On the other hand, local tissue damage resulting from envenomation may continue to progress even after treatment [11]. In a study conducted in patients with blisters formed after *Bothrops* envenomation, Gimenes et al. demonstrated the presence of antivenom and venom in blister fluids, with a higher concentration of antivenom, suggesting that there is complex formation in a local setting [11]. Therefore, the local damage observed in patients initially may be related to venom action, but later may be carried out by pro-inflammatory endogenous pathways, that are activated after envenomation [69]. This feature also highlights the importance of immediate treatment after snakebite, since this time gap may play an important role in the clinical prognosis of patients [70,71]

The authors understand the importance of identifying clinic-epidemiological variables associated to the histopathological findings, however the design of our study is a case series, with descriptive results from a limited number of cases. Our observations do not necessarily mean that dermatopathological alterations are not dependent on the characteristics mentioned above, but they may reflect a type II error likely to occur when sample sizes are too small. Although case series design is not considered the strongest source of evidence for clinical decisions, those are particularly important to provide descriptive novelty and contributes to building knowledge and generating hypotheses. The results presented here are essential to understand local injury at a clinical and histological level and serve as a basis for future studies concerned to the cause, development and structural/functional changes of the local *Bothrops* envenomation.

## Conclusion and perspectives

Tissue damage caused by *Bothrops* envenomation leads to important histopathological alterations in patients, which could be related to both direct venom activity as well as inflammatory responses resulting from envenomation, or the establishment of an infectious process. Even though the histopathological features show non-specific patterns corresponding to snakebite classification or time window of envenomation and biopsy, the persistence of findings such as hemorrhage, inflammatory infiltrate, dermal edema and congestion, even days after antivenom administration, could indicate that after the first stimuli of venom, inflammation plays a major role in the progression of tissue damage. Nonetheless, histopathological analyses of human skin lesions can enlighten the pathological, and endogenous, effects of local envenomation and possibly open ways to novel treatments, adjuvants or changes in clinical management that lead to better outcomes for patients.

## Acknowledgments

We would like to thank the Dermatology and Pathology Departments and Emergency Service at Fundação de Medicina Tropical Dr. Heitor Vieira Dourado.

## Author Contributions

**Conceptualization:** Fabiane Bianca Albuquerque Barbosa, Rima de Souza Raad, Hiochelson Najibe Santos Ibiapina, Jacqueline Sachett, Marco Aurélio Sartim, Wuelton Monteiro, Allyson Guimarães Costa, Luiz Carlos Lima Ferreira.

**Data curation:** Fabiane Bianca Albuquerque Barbosa, Rima de Souza Raad, Hiochelson Najibe Santos Ibiapina, Monique Freire dos Reis, Rosilene Viana Andrade, Thaís Pinto Nascimento, Fabio Francesconi Valle, Nicholas R. Casewell, Jacqueline Sachett, Marco Aurélio Sartim, Wuelton Monteiro, Allyson Guimarães Costa, Luiz Carlos Lima Ferreira.

**Formal analysis:** Fabiane Bianca Albuquerque Barbosa, Hiochelson Najibe Santos Ibiapina, Monique Freire dos Reis, Juliana Costa Ferreira Neves, Thaís Pinto Nascimento, Jacqueline Sachett, Marco Aurélio Sartim, Wuelton Monteiro, Allyson Guimarães Costa, Luiz Carlos Lima Ferreira.

**Funding acquisition:** Jacqueline Sachett, Wuelton Monteiro, Allyson Guimarães Costa.

**Investigation:** Fabiane Bianca Albuquerque Barbosa, Rima de Souza Raad, Hiochelson Najibe Santos Ibiapina, Monique Freire dos Reis, Fabio Francesconi Valle, Nicholas R. Casewell, Jacqueline Sachett, Marco Aurélio Sartim, Wuelton Monteiro, Allyson Guimarães Costa, Luiz Carlos Lima Ferreira.

**Methodology:** Fabiane Bianca Albuquerque Barbosa, Rima de Souza Raad, Hiochelson Najibe Santos Ibiapina, Rosilene Viana Andrade, Jacqueline Sachett, Marco Aurélio Sartim, Wuelton Monteiro, Allyson Guimarães Costa, Luiz Carlos Lima Ferreira.

**Project administration:** Fabiane Bianca Albuquerque Barbosa, Jacqueline Sachett, Wuelton Monteiro, Allyson Guimarães Costa, Luiz Carlos Lima Ferreira.

**Resources:** Jacqueline Sachett, Wuelton Monteiro, Allyson Guimarães Costa, Luiz Carlos Lima Ferreira.

**Software:** Fabiane Bianca Albuquerque Barbosa, Wuelton Monteiro, Allyson Guimarães Costa, Luiz Carlos Lima Ferreira.

**Supervision:** Fabiane Bianca Albuquerque Barbosa, Jacqueline Sachett, Marco Aurélio Sartim, Wuelton Monteiro, Allyson Guimarães Costa, Luiz Carlos Lima Ferreira.

**Validation:** Fabiane Bianca Albuquerque Barbosa, Monique Freire dos Reis, Rosilene Viana Andrade, Fabio Francesconi Valle, Nicholas R. Casewell, Jacqueline Sachett, Wuelton Monteiro, Allyson Guimarães Costa, Luiz Carlos Lima Ferreira.

**Visualization:** Fabiane Bianca Albuquerque Barbosa, Jacqueline Sachett, Marco Aurélio Sartim, Wuelton Monteiro, Allyson Guimarães Costa, Luiz Carlos Lima Ferreira.

**Writing – original draft:** Fabiane Bianca Albuquerque Barbosa, Jacqueline Sachett, Marco Aurélio Sartim, Wuelton Monteiro, Allyson Guimarães Costa, Luiz Carlos Lima Ferreira.

**Writing – review & editing:** Fabiane Bianca Albuquerque Barbosa, Hiochelson Najibe Santos Ibiapina, Monique Freire dos Reis, Juliana Costa Ferreira Neves, Rosilene Viana Andrade, Thaís Pinto Nascimento, Fabio Francesconi Valle, Nicholas R. Casewell, Jacqueline Sachett,

Marco Aurélio Sartim, Wuelton Monteiro, Allyson Guimarães Costa, Luiz Carlos Lima Ferreira.

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
