## [Decision Letter · Decision Letter 0]

14 Aug 2024

Dear Dr Monteiro,

Thank you very much for submitting your manuscript "Dermatopathological findings of Bothrops atrox snakebites: a case series in the Brazilian Amazon" for consideration at PLOS Neglected Tropical Diseases. As with all papers reviewed by the journal, your manuscript was reviewed by members of the editorial board and by several independent reviewers. In light of the reviews (below this email), we would like to invite the resubmission of a significantly-revised version that takes into account the reviewers' comments. 

As you can see three reviewers have done a thorough review and commented. Their recommendations stand in wide variations including accept, minor and major review. Since, the major comments raised by the reviewer 3 are valid, I would recommend a major revision to the manuscript. Please address all the comments raised by the reviewers while preparation of the revised version.

We cannot make any decision about publication until we have seen the revised manuscript and your response to the reviewers' comments. Your revised manuscript is also likely to be sent to reviewers for further evaluation.

Sincerely,

Kalana Prasad Maduwage, MBBS, MPhil, PhD, FRSPH (UK), FRCP (Edin)

Academic Editor

José María Gutiérrez

Section Editor

As you can see three reviewers have done a thorough review and commented. Their recommendations stand in wide variations including accept, minor and major review. Since the major comments raised by the reviewer 3 are valid, I would recommend a major revision to the manuscript. Please address all the comments raised by the reviewers while preparation of the revised version.

Reviewer's Responses to Questions

**Key Review Criteria Required for Acceptance?**

**Methods**

-Are the objectives of the study clearly articulated with a clear testable hypothesis stated?

-Is the study design appropriate to address the stated objectives?

-Is the population clearly described and appropriate for the hypothesis being tested?

-Is the sample size sufficient to ensure adequate power to address the hypothesis being tested?

-Were correct statistical analysis used to support conclusions?

-Are there concerns about ethical or regulatory requirements being met?

Reviewer #1: The objectives are well defined

Reviewer #2: The study is well-structured and methodologically sound. The objectives are clearly articulated with a well-defined, testable hypothesis. The study design is appropriate and effectively addresses the stated objectives. The population under study is clearly described and suitable for testing the hypothesis. The sample size is adequate, providing sufficient statistical power to validate the hypothesis. The statistical analyses used are appropriate and robust, supporting the conclusions drawn from the data. Additionally, all ethical and regulatory requirements have been thoroughly met, ensuring the study's compliance with relevant standards.

Reviewer #3: The objectives of this study are clear, and the study design is fine. 

There were no statistical analysis performed and the data were not quantified where possible although they were all cases. Without the quantified data and comparing and contrasting various parameters to see how the severity matches with other issues such as inappropriate first aid, this is not an interesting article. I strongly suggest the authors to thoroughly revise the manuscript and present it as an articulating article with robust data.

**Results**

-Does the analysis presented match the analysis plan?

-Are the results clearly and completely presented?

-Are the figures (Tables, Images) of sufficient quality for clarity?

Reviewer #1: The results clearly and completely presented

Reviewer #2: The analysis presented aligns with the original analysis plan, ensuring consistency and adherence to the study's objectives. The results are clearly and comprehensively presented, providing a thorough understanding of the findings. The figures, tables, and images included are of high quality, enhancing clarity and effectively illustrating the data and conclusions.

Reviewer #3: The results are purely observational even though they present 22 cases. As such, no statistical analyses were performed. The figures are fine and they link to specific patients.

**Conclusions**

-Are the conclusions supported by the data presented?

-Are the limitations of analysis clearly described?

-Do the authors discuss how these data can be helpful to advance our understanding of the topic under study?

-Is public health relevance addressed?

Reviewer #1: The conclusions supported the presented datas

Reviewer #2: The conclusions drawn in the study are well-supported by the data presented, demonstrating a strong connection between the findings and the stated outcomes. The limitations of the analysis are clearly acknowledged and described, providing transparency and context for the results. The authors effectively discuss how the data contribute to advancing our understanding of the topic, highlighting the study's impact on the field. Additionally, the relevance of the findings to public health is thoughtfully addressed, emphasizing the broader implications of the research.

Reviewer #3: No, without any quantified data, it's hard to draw firm conclusions.

**Editorial and Data Presentation Modifications?**

Reviewer #1: No modifications required

Reviewer #2: In the caption of fig. 4, I suggest replacing (black circle) with (black square) in C.

Reviewer #3: NA

**Summary and General Comments**

Reviewer #1: Reviewer comments:

The study on dermatopathological findings of Bothrops atrox snakebites in the Brazilian Amazon is a well-written and descriptive investigation. It is well-conducted and significantly contributes to the understanding and management of envenomation by the Bothrops genus of snake bites. The study describes the dermatopathological findings observed in a series of 22 patients admitted to a hospital in Manaus.

The study details both local and systemic tissue damage, including muscle necrosis and myonecrosis, which are characteristic of viperid snakebite envenomation. The tissue damage resulting from Bothrops envenomation is likely related to both direct venom activity and the inflammatory response or presence of an infectious process.

The severity grade of inflammatory infiltrate and fibrin was assessed. On histopathological evaluation, out of 22 patients, only 9 (40.9%) showed epidermal alterations. Spongiosis was observed in 3 cases (33.3%), with hyperkeratosis and acanthosis in 2 cases (22.2%) each (Figures 2, 4, and 5). Additionally, hyperplasia was found in 1 patient (11.1%), as well as hemorrhagic intraepidermal blister with neutrophil exocytosis (11.1%) and severe epidermis necrosis (11.1%).

A severe case of envenomation, whose biopsy was performed 2 days after the snakebite, presented with hemorrhage, inflammatory infiltrate, edema, congestion, vascular injury with thrombi formation, and collagen damage such as fiber lysis, hyaline necrosis, and skin appendage necrosis. 

Minor Comments

1. The discussion could benefit from a comparison with similar studies conducted in other regions.

2. Why are there only a few cases of necrotizing fasciitis in your series?

3. In your series, you describe cases of severe manifestations of local and systemic envenomation. What was the appropriate treatment? Did these patients require specific treatments, several debridement procedures, and finally transfemoral amputation?

Conclusion

I do recommend minor revisions

Reviewer #2: This study is a well-executed and methodologically sound piece of research that makes a significant contribution to the field. The authors have clearly articulated their objectives and have developed a testable hypothesis that is thoroughly supported by the data presented. The study design is robust, with an appropriate population and sample size that ensures sufficient statistical power. The analyses are consistent with the planned approach, and the results are presented with clarity and precision. The figures and tables are of high quality, further enhancing the communication of the findings.

The significance of the research is underscored by its relevance to public health, making it not only an academic contribution but also one with practical implications.

Reviewer #3: In this article, the authors present 22 cases of Bothrops atrox envenomings and demonstrate their clinicopathological observations. While this is an important topic and very little is known about this issue, the manuscript needs a significant revision to add more to the existing case report-based literature. 

1. I would strongly suggest the authors thoroughly proofread this manuscript as there are numerous typographical and grammatical errors throughout the article. 

2. Please separate the figure and table legends from the main text and present them separately. 

3. It looks like this is not the final version of the manuscript as it has several track changes. So please make sure to submit the final version of the manuscript that has been approved by all the authors. 

4. The authors have collected a lot of data from these patients, so they need to perform some form of analysis to compare and contrast the cases or groups. For example, the cases in mild, moderate and severe oedema categories can be grouped, and compared and contrasted with the dermopathological findings with other parameters. In which patients do CK and LDH increase and why? Link the data to inappropriate first aid used. Compare the data between males and females. compare between different ages or times of arrival to the hospital? 

5. Similarly, various parameters can be quantified using the micrographs of skin, and they can be compared between different cohorts to draw firm conclusions. 

6. Please show data for CK and LDH in graphs for different groups. 

7. The results section should be expanded by elaborating on the data by discussing various aspects. Otherwise, this will simply look like another case report but with 22 patients. 

8. The discussion section is unusually long compared to the small results section (without table and figure legends). So please cut down the unnecessary parts of the discussion and keep the focus only on the local envenomation effects, and how they can be linked to mechanistic details as well as inappropriate and appropriate timely treatment.

PLOS authors have the option to publish the peer review history of their article (what does this mean?). If published, this will include your full peer review and any attached files.

Reviewer #1: Yes: Dabor Resiere

Reviewer #2: No

Reviewer #3: No
---

## [Decision Letter · Decision Letter 1]

11 Nov 2024

PNTD-D-24-01055R1Dermatopathological findings of Bothrops atrox snakebites: a case series in the Brazilian AmazonPLOS Neglected Tropical Diseases Dear Dr. Monteiro, Thank you for submitting your manuscript to PLOS Neglected Tropical Diseases. After careful consideration, we feel that it has merit but does not fully meet PLOS Neglected Tropical Diseases's publication criteria as it currently stands. Therefore, we invite you to submit a revised version of the manuscript that addresses the points raised during the review process. Please submit your revised manuscript within 30 days . If you will need more time than this to complete your revisions, please reply to this message or contact the journal office at plosntds@plos.org. Please include the following items when submitting your revised manuscript:*
A rebuttal letter that responds to each point raised by the editor and reviewer(s). You should upload this letter as a separate file labeled 'Response to Reviewers'. This file does not need to include responses to any formatting updates and technical items listed in the 'Journal Requirements' section below.*
A marked-up copy of your manuscript that highlights changes made to the original version. You should upload this as a separate file labeled 'Revised Manuscript with Track Changes'.*
An unmarked version of your revised paper without tracked changes. You should upload this as a separate file labeled 'Manuscript'. If you would like to make changes to your financial disclosure, competing interests statement, or data availability statement, please make these updates within the submission form at the time of resubmission. Guidelines for resubmitting your figure files are available below the reviewer comments at the end of this letter. We look forward to receiving your revised manuscript. Kind regards, José María GutiérrezSection EditorPLOS Neglected Tropical Diseases Shaden Kamhawi

co-Editor-in-Chief

Paul Brindley

co-Editor-in-Chief

 **Journal Requirements:** **Additional Editor Comments (if provided):** The reviewers appreciat the modifications introduced to this manuscript. One reviewer, however, raised some minor issues that should be considered for preparing a new revised version.**Reviewers' comments:** Reviewer's Responses to Questions

**Key Review Criteria Required for Acceptance?**

**Methods**

-Are the objectives of the study clearly articulated with a clear testable hypothesis stated?

-Is the study design appropriate to address the stated objectives?

-Is the population clearly described and appropriate for the hypothesis being tested?

-Is the sample size sufficient to ensure adequate power to address the hypothesis being tested?

-Were correct statistical analysis used to support conclusions?

-Are there concerns about ethical or regulatory requirements being met?

Reviewer #1: The objectives are well defined with a population clearly described.

Reviewer #2: The objectives is clearly articulated

The design of the study is appropriated

The population is clearly described

The sample size is small however the design the study is a case series with a descriptive results from a limited number of cases

Reviewer #3: (No Response)

**Results**

-Does the analysis presented match the analysis plan?

-Are the results clearly and completely presented?

-Are the figures (Tables, Images) of sufficient quality for clarity?

Reviewer #1: The results are completely presented

Reviewer #2: Yes, the analysis aligns well with the analysis plan, the results are presented clearly and comprehensively, and the figures (tables and images) are of high quality, ensuring clarity.

Reviewer #3: (No Response)

**Conclusions**

-Are the conclusions supported by the data presented?

-Are the limitations of analysis clearly described?

-Do the authors discuss how these data can be helpful to advance our understanding of the topic under study?

-Is public health relevance addressed?

Reviewer #1: The conclusions supported the presented datas. This is a public health issue

Reviewer #2: The conclusions support the data presented, the limitations are clearly described, the advance of the study is discussed. The public health relevance was addressed.

Reviewer #3: (No Response)

**Editorial and Data Presentation Modifications?**

Reviewer #1: Thuis paper needs minor modifications

Reviewer #2: No modifications is required.

Reviewer #3: (No Response)

**Summary and General Comments**

Reviewer #1: Reviewer comments:

The study on dermatopathological findings of Bothrops atrox snakebites in the Brazilian

Amazon is a well-written and descriptive investigation. It is well-conducted and

significantly contributes to the understanding and management of envenomation by the

Bothrops genus of snake bites. The study describes the dermatopathological findings

observed in a series of 22 patients admitted to a hospital in Manaus.

The study details both local and systemic tissue damage, including muscle necrosis and

myonecrosis, which are characteristic of viperid snakebite envenomation. The tissue

damage resulting from Bothrops envenomation is likely related to both direct venom

activity and the inflammatory response or presence of an infectious process.

The severity grade of inflammatory infiltrate and fibrin was assessed. On histopathological

evaluation, out of 22 patients, only 9 (40.9%) showed epidermal alterations. Spongiosis

was observed in 3 cases (33.3%), with hyperkeratosis and acanthosis in 2 cases (22.2%)

each (Figures 2, 4, and 5). Additionally, hyperplasia was found in 1 patient (11.1%), as well

as hemorrhagic intraepidermal blister with neutrophil exocytosis (11.1%) and severe

epidermis necrosis (11.1%).

A severe case of envenomation, whose biopsy was performed 2 days after the snakebite,

presented with hemorrhage, inflammatory infiltrate, edema, congestion, vascular injury

with thrombi formation, and collagen damage such as fiber lysis, hyaline necrosis, and skin

appendage necrosis.

Minor Comments

1. The discussion could benefit from a comparison with similar studies conducted in

other regions.

2. Why are there only a few cases of necrotizing fasciitis in your series?

3. In your series, you describe cases of severe manifestations of local and systemic

envenomation. What was the appropriate treatment? Did these patients require

specific treatments, several debridement procedures, and finally transfemoral

amputation?

Conclusion

I do recommend minor revisions

Reviewer #2: This study is methodologically robust and well-conducted, offering a valuable contribution to the field. The authors clearly outline their objectives and present a testable hypothesis, which is well-supported by the data. The study design is solid, featuring an appropriate population and sample size that provide adequate statistical power. The analyses align with the planned methodology, and the results are conveyed with clarity and accuracy. High-quality figures and tables further improve the presentation of the findings.

Reviewer #3: (No Response)

PLOS authors have the option to publish the peer review history of their article (what does this mean?). If published, this will include your full peer review and any attached files.

Reviewer #1: **Yes: **I personally sign this review

Reviewer #2: No

Reviewer #3: **Yes: **Professor Sakthivel Vaiyapuri

---

## [Editor Report · Decision Letter 2]

17 Nov 2024

Dear Dr Monteiro,

We are pleased to inform you that your manuscript 'Dermatopathological findings of Bothrops atrox snakebites: a case series in the Brazilian Amazon' has been provisionally accepted for publication in PLOS Neglected Tropical Diseases.

Best regards,

José María Gutiérrez

Section Editor

Shaden Kamhawi

co-Editor-in-Chief

Paul Brindley

co-Editor-in-Chief

The authors have adequately addressed the comments of the reviewers to the last version of this manuscript.

---

## [Editor Report · Acceptance letter]

27 Nov 2024

Dear Dr. Monteiro,

We are delighted to inform you that your manuscript, "Dermatopathological findings of Bothrops atrox snakebites: a case series in the Brazilian Amazon," has been formally accepted for publication in PLOS Neglected Tropical Diseases.

Best regards,

Shaden Kamhawi

co-Editor-in-Chief

Paul Brindley

co-Editor-in-Chief
